# Three-Dimensional Reconstruction of Indoor Scenes Based on Implicit Neural Representation

**DOI:** 10.3390/jimaging10090231

**Published:** 2024-09-16

**Authors:** Zhaoji Lin, Yutao Huang, Li Yao

**Affiliations:** 1School of Computer Science and Engineering, Sanjiang University, Nanjing 210012, China; 2School of Computer Science and Engineering, Southeast University, Nanjing 211189, China; 172210501215@stu.just.edu.cn (Y.H.); yao.li@seu.edu.cn (L.Y.)

**Keywords:** 3D reconstruction, indoor scene, neural radiance fields, signed distance function, normal prior, mesh

## Abstract

Reconstructing 3D indoor scenes from 2D images has always been an important task in computer vision and graphics applications. For indoor scenes, traditional 3D reconstruction methods have problems such as missing surface details, poor reconstruction of large plane textures and uneven illumination areas, and many wrongly reconstructed floating debris noises in the reconstructed models. This paper proposes a 3D reconstruction method for indoor scenes that combines neural radiation field (NeRFs) and signed distance function (SDF) implicit expressions. The volume density of the NeRF is used to provide geometric information for the SDF field, and the learning of geometric shapes and surfaces is strengthened by adding an adaptive normal prior optimization learning process. It not only preserves the high-quality geometric information of the NeRF, but also uses the SDF to generate an explicit mesh with a smooth surface, significantly improving the reconstruction quality of large plane textures and uneven illumination areas in indoor scenes. At the same time, a new regularization term is designed to constrain the weight distribution, making it an ideal unimodal compact distribution, thereby alleviating the problem of uneven density distribution and achieving the effect of floating debris removal in the final model. Experiments show that the 3D reconstruction effect of this paper on ScanNet, Hypersim, and Replica datasets outperforms the state-of-the-art methods.

## 1. Introduction

The goal of 3D reconstruction of indoor scenes is to reconstruct and restore an accurate scene model from 2D images of indoor scenes from multiple angles to reflect the geometry, structure, and appearance characteristics of the actual scene [1]. This process can obtain an explicit 3D model observed from any perspective. This task has been a hot topic and an important task in computer vision and graphics research in recent years, and has broad application prospects in house interior restoration, interior design, virtual reality, augmented reality, indoor navigation, etc. [2].

Unlike object-level reconstruction, indoor environments usually include large and small objects, different materials, and complex spatial layouts, which puts higher demands on feature extraction and scene understanding. Indoor lighting conditions are also complex and changeable, and may include natural light, artificial light, and shadow areas. These factors will affect the quality of the reconstructed model and the difficulty of reconstruction. In addition, occlusion between objects is more common in indoor scenes, which makes it more difficult to obtain complete scene information from a limited perspective. Existing methods often have poor effects on uneven lighting (as shown in Figure 1a) and partial planar texture processing (as shown in Figure 1b). There are many floating debris noises in the air of the reconstructed indoor 3D model (as shown in Figure 1c). The recently proposed neural radiance fields reconstruction method can achieve good results, but the computation is large, and it cannot directly obtain a 3D mesh model. We strengthen surface learning by adding normal priors and use an SDF, a compact and continuous multi-layer perceptron (MLP), to parameterize the representation of the implicit model, and finally obtain a high-quality 3D mesh model. In summary, this paper has the following main contributions:(1)It proposes a new indoor 3D reconstruction method that combines NeRF and SDF scene expression, which not only preserves the high-quality geometric information of the NeRF, but also uses the SDF to generate an explicit mesh with a smooth surface.(2)By adding adaptive normal priors to provide globally consistent geometric constraints, the reconstruction quality of planar texture areas and details is significantly improved.(3)By introducing a new regularization term, the problem of uneven distribution of NeRF density is alleviated, and the effect of removing floating debris is achieved in the final generated model, which improves the look and feel of the visualization results.

## 2. Related Works

### 2.1. Three-Dimensional Reconstruction of Based on Visual SLAM

Simultaneous Localization and Mapping (SLAM) refers to the process of a moving object carrying a sensor to locate itself during movement and to synchronously map the surrounding environment in an appropriate manner. In 2016, Google open-sourced an indoor SLAM library called Cartographer [3], which is still being updated. Its main application area is indoor reconstruction, providing functions such as positioning, mapping, and loop detection. LOAM [4] is a SLAM algorithm based on a 3D laser sensor. Compared with Cartographer, it can solve indoor and outdoor problems, but it does not have loop detection. Later, based on LOAM, researchers proposed LeGOLOAM [5], a new algorithm derived from the LOAM framework. By introducing a global map and loop detection module, the positioning accuracy and robustness are improved, and at the same time, the entire algorithm is made more lightweight. The advantage of the SLAM system is its fast reconstruction speed. With the improvement of the robustness and accuracy of its algorithm, it makes real-time 3D reconstruction possible. However, due to the need for sensor participation, the reconstruction cost is high, and the process is relatively complicated.

### 2.2. Three-Dimensional Reconstruction Based on TSDF

The main idea of the 3D reconstruction method based on regression to a truncated signed distance function (TSDF) is to create a voxel grid, initialize the maximum truncated signed distance value for each voxel, fuse the depth map into the voxel through the TSDF [6], calculate the distance from each point to the center of each voxel in the voxel grid, and update the TSDF value of the voxel if the distance is within the truncation range. KinectFusion [7] uses the depth camera Kinect to scan and model indoor spaces and objects in real time, and uses a TSDF to manage spatial information. This method improves the efficiency of data processing and the accuracy of the reconstruction process. Atlas [8] provides an end-to-end reconstruction pipeline, using a 2D CNN to extract features from each image independently, and then using the intrinsic and extrinsic features of the camera to back-project and accumulate into voxel volumes. After accumulation, a 3D CNN refines the accumulated features and predicts the TSDF value. At the same time, the semantic segmentation goal is added to the model to accurately mark the generated surface, which improves the model’s ability to handle occlusion and large room scenes. In order to reduce the computational burden, unlike Atlas, which processes the entire image sequence at once, NeuralRecon [9] proposed a coarse-to-fine framework that uses a recursive network to fuse features from previous fragments and reconstructs the entire scene by processing the local surface of each fragment sequence, making progress on datasets with high occlusion and scene complexity. However, due to its design idea of local estimation, TSDF-based 3D reconstruction methods have difficulty obtaining global reconstruction with fine details.

### 2.3. Three-Dimensional Reconstruction Based on MVS

The traditional multi-view stereo (MVS) method first estimates the depth map of each image based on multiple views, and then performs depth fusion to obtain the final reconstruction result. These methods can reconstruct relatively accurate 3D shapes and have been used in many downstream applications, such as new view synthesis. Schoenberger proposed a new general image 3D reconstruction method, which uses scale-invariant feature transform (SIFT) features and the Fast Library for Approximate Nearest Neighbors (FLANN) matching algorithm to improve the accuracy and efficiency of Structure from Motion (SFM), and open-sourced the project as COLMAP 3.10 [10] software for enthusiasts to use, which can perform dense point cloud reconstruction and surface reconstruction. Based on this, they further proposed the Pixelwise View Selection [11] method, which improved the poor reconstruction effect and efficiency caused by the previous input image specifications (such as different input image sizes, different lighting, etc.). This method regards the view selection problem as a binary classification problem, selects the best view for each pixel, and thus can use the information of all perspectives for better stereo matching. However, these methods perform poorly in large planar texture areas or areas with sparse textures because their optimization is highly dependent on the photometric process. In indoor scenes with planar texture areas, the inherent uniformity makes photometry ineffective, making it difficult to accurately estimate depth [12].

With the development of deep learning, learning-based MVS methods [13,14,15] have shown good performance in recent years. Mvsnet [16] uses convolutional neural networks to predict depth maps, integrates image information from multiple perspectives into a unified 3D space, and implements depth estimation by constructing a 3D matching cost-volume, which significantly improves the accuracy of depth estimation. Fast-Mvsnet [17] improves on Mvsnet by adopting a more efficient network structure and optimizing the depth map prediction process. By simplifying the representation of cost volume and adopting a lighter network architecture, the processing speed is improved, and the memory consumption is reduced. DeepV2D [18] combines the temporal information of video frames with visual depth estimation and adopts a two-stage network architecture. It first performs motion estimation on the video sequence, and then combines motion information and visual features to predict the depth map of each frame, further improving the quality of depth prediction. DeepMVS [19] uses a deep neural network to preprocess the input image, extract features, and predict the depth information of each pixel. It introduces a novel image warping and synthesis step to improve the consistency between views. UCS-Net [20] constructs the cost volume in a coarse-to-fine hierarchical manner to obtain higher resolution depth estimation. Although the MVS method based on deep learning has made great progress, it still faces some problems. Since the depth map is estimated separately for each view, there are often some geometric inconsistencies and scale ambiguities, resulting in holes and noise on the surface of the reconstructed result [21]. When encountering areas with relatively scarce textures, these models find it difficult to accurately predict depth information.

### 2.4. Three-Dimensional Reconstruction Based on Implicit Neural Networks 

Coordinate-based implicit neural networks, which encode a field by regressing 3D coordinates to output values via an MLP, have become a popular approach for representing scenes due to their compactness and flexibility. The Dist [22] model proposed a method to learn 3D shapes from 2D images. IDR [23] models rely on view appearance and can be applied to non-Lambertian surface reconstruction. However, they require mask information to obtain reconstructions. NeRFs encode scene geometry via volume density and are suitable for the task of novel view synthesis for volume rendering. However, due to the lack of surface constraints, volume density cannot represent high-fidelity surfaces. Inspired by neural radiance fields, Neus [24] and VolSDF [25] attached volume rendering techniques to IDR and eliminated the need for mask information. Although these methods achieve stunning reconstruction results in small-scale, texture-rich scenes, they often perform poorly in large-scale indoor scenes with planar texture areas. Mip-NeRF360 [26] improves upon the original NeRF’s problems of unbalanced details and proportions at near and far distances, as well as the limited nature of synthesized scenes, and can render unbounded scenes more realistically. To address the slow convergence of the original NeRF training process, NSVF [27] uses a sparse voxel octree to assist in spatial modeling, achieving significant improvements in training time. The traditional NeRF requires input from multiple views to estimate volume representations. If multi-view data are insufficient, the generated scene can easily collapse into a plane. To address this problem, Google researchers proposed LOLNeRF [28], which can train a NeRF model from a single viewpoint for the same type of object, without adversarial supervision, thereby enabling a single 2D image to generate a 3D model. All of the above are advances made by NeRFs in synthesizing new viewpoints. In recent years, researchers have gradually shifted their focus to using NeRFs for 3D reconstruction. Guy [29] et al. applied a NeRF to facial reconstruction, achieving the generation of high-quality 3D facial models from a single RGB image. By introducing the time dimension, this method can model and reconstruct the dynamic changes in facial shape and expression. However, all of the above NeRF studies are limited to object-level reconstruction. When the reconstructed objects are expanded to indoor scenes, the generated 3D models often contain a lot of noise and topological errors.

## 3. Methodology

This paper proposes a method for indoor scene 3D reconstruction that combines NeRF and SDF implicit expression. The volume density of the NeRF is used to provide geometric information for the SDF field, and the learning process is optimized by adding normal priors to strengthen the learning of geometric shapes and surfaces. The overall framework of the method proposed in this paper is shown in Figure 2. Multiple 2D images of indoor scenes are used as input, and the explicit 3D model mesh of the corresponding scene is output. This method mainly includes the following three modules:Normal estimation module: This module uses a spatial rectifier-based method to generate the corresponding normal map for a single RGB image, and prepares data for the prior part of neural implicit reconstruction.NeRF module: The appearance decomposition and feature processing of the scene image are performed through the neural radiant field, and the volume density and color are obtained. The image under the corresponding perspective is obtained by volume rendering, and the MLP parameters are optimized inversely with the input image loss.SDF field module: The purpose of this module is to learn a high-quality SDF from the network, and at the same time, strengthen the network’s understanding of the geometric structure through the normal prior. The implicit-3D-expression SDF is converted into an explicit triangular mesh through the Marching Cubes algorithm.

### 3.1. Optimization of Indoor 3D Reconstruction Based on Adaptive Normal Prior

A normal map is an important type of image, which represents the surface normal direction of each pixel in the image through the color of the pixel. In 3D graphics and computer vision, the normal is a vector perpendicular to the surface of an object, which can be used to describe the direction and shape of the surface. In the normal map, RGB color channels are generally used to represent the X, Y, and Z components of the normal vector, respectively. Normal information helps to correct errors in the reconstruction process and improve the reconstruction quality.

Currently, many monocular normal estimation methods have achieved high accuracy under a clear image input. However, considering that indoor scene images often have small blur and tilt, this paper selects TiltedSN [30] as the normal estimation module. Because the estimated normal map is usually over-smoothed, there are problems of inaccurate estimation on some fine structures, such as the chair legs in Figure 3a. Therefore, we adopt an adaptive method to use normal priors, using a mechanism based on the consistency of multiple views of the input image to evaluate the reliability of the normal prior. As shown in Figure 3b, this process is also called geometric checking. For areas that do not meet the consistency of multiple images, the normal prior is not applied. Instead, the appearance information is used for optimization to avoid the negative effects of incorrect normal maps that lead to misjudgment in reconstruction.

Given a reference image $I_{i}$, evaluate the consistency of the surface observed from pixel q. Define a local 3D plane $\left\{p|p^{T}n=dv^{T}n\right\}\;\;$ in the camera space associated with q, where v is the viewing direction, d is the distance to pixel q, and n is the normal estimate. Next, find a set of adjacent images, assuming that one of the adjacent images is $I_{j}$. The homography change from $I_{i}$ to $I_{j}$ can be calculated by the following formula:(1)$$H_{n,d}=K_{j}\left(R_{j}R_{i}^{-1}-\frac{\left(t_{i}-t_{j}\right)n^{T}}{dv^{T}n}\right)K_{i}^{-1}$$
where $\left\{K_{*},\;R_{*},\;t_{*}\right\}$ is the intrinsic parameter matrix, the rotation and translation camera parameters. For pixel q in $I_{i}$, find a square block P centered on it as the neighborhood, and warp the block to its adjacent view $I_{j}$ using the calculated homography matrix. The block matching method (patchmatch) can be used to find similar image blocks on adjacent views, and the normalized cross-correlation (NCC) method is used to evaluate the visual consistency of (n, d). NCC is a method for measuring the similarity between two images. It evaluates the similarity between the two images by calculating the degree of correlation between them. Compared with simple cross-correlation, normalized cross-correlation is insensitive to changes in brightness and contrast, so it is more reliable in practical applications. The applied NCC formula is as follows:(2)$$NCC_{j}\left(P,n\right)=\frac{\sum_{q\in P}{\hat{I}}_{i}\left(q\right){\hat{I}}_{j}\left(H_{n,d}\left(q\right)\right)}{\sqrt{\sum_{q\in P}{\hat{I}}_{i}{\left(q\right)}^{2}\sum_{q\in P}{\hat{I}}_{j}{\left(H_{n,d}\left(q\right)\right)}^{2}}}$$
where ${\hat{I}}_{*}\left(q\right)=I_{*}\left(q\right)-{\bar{I}}_{*}\left(q\right)$, $I_{i}\left(q\right)$ and $I_{j}\left(q\right)$ represent two image regions to be compared, ${\bar{I}}_{*}\left(q\right)$ is the average value of $I_{*}\left(q\right)$, ${\hat{I}}_{*}\left(q\right)$ is the difference between them; the numerator calculates the sum of the products of the differences between the two image blocks, and the denominator calculates the square root of the product of the sum of the squares of the differences between the two image blocks. This process ensures the normalization of the results, making the NCC range within (−1, 1). The closer the NCC value is to 1, the more similar the two image regions are; the closer the NCC value is to −1, the less similar they are; the closer the NCC value is to 0, the less obvious linear relationship there is between them.

If the reconstructed geometry is not accurate at the sampling pixel, it cannot meet the multi-view photometric consistency, which means that its related normal prior cannot provide help for the overall reconstruction. Therefore, a threshold ϵ is set, and by comparing the NCC at the sampling block with ϵ, the following indicator function can be used to adaptively determine the training weight of the normal prior.
(3)$$\Omega_{q}\left(\hat{n}\right)=\left\{\begin{array}{c} 1\;\;\;\;if\sum_{j}NCC_{j}\left(P,\hat{n}\right)\geq \epsilon \\ 0\;\;\;\;if\sum_{j}NCC_{j}\left(P,\hat{n}\right)<\epsilon \end{array}\right.$$

The normal prior is used for supervision only when $\Omega_{q}\left(\hat{n}\right)=1$. If $\Omega_{q}\left(\hat{n}\right)=0$, the normal prior of the region is considered unreliable and will not be used in subsequent optimization processes.

### 3.2. Neural Implicit Reconstruction

Since the NeRF cannot express the surface of the object well, we are looking for a new implicit expression. The SDF can represent the surface information of objects and scenes and achieve better surface reconstruction. Therefore, we choose the SDF as the implicit scene representation.

#### 3.2.1. Scene Representation

We represent the scene geometry as an SDF. An SDF is a continuous function f that, for a given 3D point, returns the distance from that point to the nearest surface:(4)f:ℝ3→ℝ      x↦s=fx

Here, x represents a 3D point and s is the corresponding SDF value, thus completing the mapping from a 3D point to a signed distance. We define the surface S as the zero-level set of the SDF, expressed as follows:(5)$$S=\left\{x|f\left(x\right)=0\right\}$$

By using the Marching Cubes algorithm on the zero horizontal plane of the SDF, that is, the surface of the object or scene, we can obtain a 3D mesh with a relatively smooth surface.

Using the SDF to get the mesh has the following advantages:(1)Clear surface definition: The SDF provides the distance to the nearest surface for each spatial point, where the surface is defined as the location where the SDF value is zero. This representation is well suited for extracting clear and precise surfaces, making the conversion from SDF to mesh relatively direct and efficient.(2)Geometric accuracy: The SDF can accurately represent sharp edges and complex topological structures, which can be maintained when converted to meshes, thereby generating high-quality 3D models.(3)Optimization-friendly: The information provided by the SDF can be directly used to perform geometry optimization and mesh smoothing operations, which helps to further improve the quality of the model when generating the mesh.

#### 3.2.2. Implicit Indoor 3D Reconstruction Based on Normal Prior

The reconstruction process based on the normal prior is shown in Figure 4. The input mainly consists of three parts.

The first part is a five-dimensional vector representing the information of the sampling point $\left(x,y,z,\theta ,\phi \right)$. The second part is the volume density we obtained previously through the NeRF σ. Given that we use the SDF as a surface expression, the volume density obtained by the NeRF can represent more comprehensive scene geometry information. Because the scene contains multiple objects and air, it is difficult to represent a complex scene only through surface information. Traditional SDF-based methods are often limited to object reconstruction. The constraints of NeRF volume density combined with SDF reconstruction can reconstruct richer scene geometry. The third part is the normal geometry prior. The normal map here is obtained by the monocular normal estimation method mentioned earlier. The normal map provides the orientation information of the object surface, which is conducive to enhancing detail reconstruction.

The network used here is an improved NeRF, which also contains 2 MLPs. Like the NeRF, there is a color network, $f_{c}$, and the other grid has become an SDF network, $f_{\theta_{g}}$, which can get the SDF value of the point through the 3D coordinates of the point.

Here, we will use volume rendering technology to get the predicted image, and optimize it with the input real image through the loss function. Specifically, for each pixel, we sample a set of points along the corresponding emission light, denoted as $p_{i}=o+d_{i}v$, where $p_{i}\;$ is the sampling point, o is the camera center, and v is the direction of the light. The color value can be accumulated by the following volume rendering formula.
(6)$$\hat{c}=\sum_{i=1}^{n}T_{i}\alpha_{i}c\left(p_{i},v\right)$$
where $T_{i}=\overset{i-1}{\underset{j=1}{\prod }}\left(1-\alpha_{j}\right)$ is the cumulative transmittance, i.e., the probability that with no object occlusion, c is the color value, $\alpha_{i}=1-\exp\left(-\int_{t_{i}}^{t_{i+1}}\rho \left(t\right)dt\right)$ is the discrete opacity, and $\rho \left(t\right)$ is the opacity corresponding to the volume density σ in the original NeRF. Since the rendering process is fully differentiable, we learn the weights of $f_{c}$ and $f_{\theta_{g}}$ by minimizing the difference between the rendering result and the reference image.

In addition to generating the appearance, the above volume rendering scheme can also obtain the normal vector. We can approximate the surface normal observed from a viewpoint by volume accumulation along this ray:(7)$$\hat{n}=\sum_{i=1}^{n}T_{i}\alpha_{i}n_{i}$$
where $n_{i}=\nabla f\left(p_{i}\right)$ is the gradient at point $p_{i}$.

At this time, we compare the true normal map obtained by the monocular method with the estimated normal map obtained by the volume rendering process, and further optimize the parameters of the MLP by calculating the loss to obtain a more accurate normal map and geometric structure.

### 3.3. Floating Debris Removal

The NeRF initially acts on the generation of new perspectives on objects, maintaining a high degree of clarity and realism. This is mainly due to volume rendering. However, when the NeRF is used for 3D reconstruction, some floating debris in the air often appears. This floating debris refers to small disconnected areas in the volume space and translucent substances floating in the air. In view synthesis work, this floating debris is often not easy to detect. However, if an explicit 3D model needs to be generated, this floating debris will seriously affect the quality and appearance of the 3D model. Therefore, it is very necessary to remove the floating debris in these incorrectly reconstructed places.

This floating debris often does not appear in object reconstruction. However, in scene-level reconstruction, due to the significant increase in environmental complexity and the lack of relevant constraints on the NeRF, there is a phenomenon of inaccurate local area density prediction. Therefore, this paper proposes a new regularization term to constrain the weight distribution of the NeRF.

First, simulate the sampling process of the NeRF. In a scene, assume that there are only rigid objects, excluding the existence of translucent objects. After a ray is shot out, there will be countless sampling points on this ray. The weight value of the sampling point before the ray encounters the object should be extremely low (close to 0). When the ray contacts the rigid object, the weight value here should soar, much higher than other values. After the object, the weight value returns to a lower range. This is the desired weight distribution in an ideal state, which is a relatively compact unimodal distribution. This method defines a regularization term, which is a step function defined by a set of standardized ray distances s and the weight w after parameterizing each ray:(8)$$L_{dist}\left(s,w\right)=\iint_{-\infty }^{\infty }w_{s}\left(u\right)w_{s}\left(v\right)\left|u-v\right|d_{u}d_{v}$$

Here, u and v refer to points on the sampling ray, that is, points on the x-axis in the weight distribution diagram, $\left|u-v\right|$ is the distance between the two points, and $w_{s}\left(u\right)$ and $w_{s}\left(v\right)$ are the weight values at point u and point v, respectively. Since all particle combinations from negative infinity to positive infinity need to be exhausted, integration is performed in the front. If you want to make the loss function as small as possible, there are mainly two situations:

(1)If the distance between point u and point v is relatively far, that is, the value of $\left|u-v\right|$ is large, if you want to ensure that the value of $L_{dist}\left(s,w\right)$ is as small as possible, then either $w_{s}\left(u\right)$ or $w_{s}\left(v\right)$ needs to be small and close to zero. That is, as shown in Figure 5, (A, B), (B, D), (A, D), etc. all satisfy that $\left|u-v\right|\;$ is large and the weight value of at least one point is extremely small (close to zero);(2)If the values of $w_{s}\left(u\right)$ and $w_{s}\left(v\right)$ are both large, if you want to ensure that the value of $L_{dist}\left(s,w\right)$ is as small as possible, then the value of $\left|u-v\right|$ needs to be small, that is, the distance between points u and v is very close. As shown in Figure 5, only the combination of (B, C) satisfies the condition that the values of $w_{s}\left(u\right)$ and $w_{s}\left(v\right)$ are both large, and at this time, points u and v just meet the condition that the distance is very close.

Therefore, through the analysis of the above two cases, it can be found that the properties of the regularization term can constrain the density distribution to the ideal single-peak distribution with the good central tendency proposed before. The purpose of this regularization term is to minimize the sum of the normalized weighted absolute values of all samples along the ray, and encourage each ray to be as compact as possible, which is specifically reflected in the following steps:Minimize the width of each interval;Bring the intervals that are far apart closer to each other;Make the weight distribution more concentrated.

The regularization term above cannot be used directly for calculation because it is in integral form. In order to facilitate calculation and use, it is discretized as the following:$$L_{dist}\left(s,w\right)=\sum_{i,j}w_{i}w_{j}\left|\frac{s_{i}+s_{i+1}}{2}-\frac{s_{j}+s_{j+1}}{2}\right|+\frac{1}{3}\sum_{i}w_{i}^{2}\left(s_{i+1}-s_{i}\right)$$

This discretized form also provides a more intuitive understanding of the behavior represented by this regularization, where the first term minimizes the weighted distance between points in all intervals, and the second term minimizes the weighted size of each individual interval.

### 3.4. Training and Loss Function

During the training phase, we sample a batch of pixels and adaptively minimize the difference between the color and normal estimates and the true normal map. We sample m pixels $\left\{q_{k}\right\}\;$ and their corresponding reference colors $\left\{I\left(q_{k}\right)\right\}$ and normals $\left\{N\left(q_{k}\right)\right\}$ in each iteration. For each pixel, we sample n points in the world coordinate system along the corresponding ray, and the total loss is defined as
(9)$$L=\lambda_{c}L_{c}+\lambda_{n}L_{n}+\lambda_{e}L_{eik}+\lambda_{d}L_{dist}$$

Among them, $\lambda_{c}$, $\lambda_{n}$, $\lambda_{e}$, and $\lambda_{d}$ are the hyperparameters of color loss, normal loss, Eikonal loss, and distortion loss, respectively.

Color loss, $L_{c}$, is used to measure the difference in color between the reconstructed image and the real image:(10)$$L_{c}=\frac{1}{m}\sum_{k}∥I\left(q_{k}\right)-\hat{c}\left(q_{k}\right)∥$$

In the training phase, a batch of pixels need to be sampled. Each iteration samples m pixels {$q_{k}$} and the corresponding reference color {$I\left(q_{k}\right)$}, where $\hat{c}\left(q_{k}\right)$ represents the pixel color predicted by volume rendering.

The normal prior loss $L_{n}$ is to render the reconstructed 3D mesh of the indoor scene as a normal map, and compare it with the real normal map generated by the monocular method to obtain a loss:(11)$$L_{n}=\sum_{k}∥N\left(q_{k}\right)-\hat{n}\left(q_{k}\right){∥}_{1}\cdot \Omega_{q_{k}}\left(\hat{n}\left(q_{k}\right)\right)$$
where $N\left(q_{k}\right)$ is the true normal information, and $\hat{n}\left(q_{k}\right)$ is the normal information predicted by the gradient. $\Omega_{q_{k}}\left(\hat{n}\left(q_{k}\right)\right)$ is an indicator function used to judge the accuracy of the normal prior. Here, some data with inaccurate normal estimation are eliminated. The normal loss is mainly calculated by cosine similarity, because the direction of the normal vector is more important than its length. Cosine similarity is a good measure of the similarity of the two vectors in direction.

The Eikonal loss $L_{eik}$ [31] of the regularized SDF is
(12)$$L_{eik}=\sum_{x\in Χ}(∥\nabla f_{\theta }\left(x\right){∥}_{2}-1{)}^{2}$$
where Χ is a set of sampling points in the 3D space and the area near the surface, and x represents one of the sampling points. The reason why the gradient $\nabla f_{\theta }\left(x\right)$ needs to be close to 1 is that the ideal SDF represents the shortest distance from the current point to the surface, so the gradient direction is the direction in which the distance field change is the steepest. Assuming that x moves toward the surface along this direction by ∆d, the SDF should also change by ∆d. Therefore, in the ideal state, $\nabla f_{\theta }\left(x\right)=\partial D\left(x\right)=\Delta D\left(x\right)/\Delta x=\Delta d/\Delta d=1$, where $D\left(x\right)$ is the original Eikonal equation. By introducing the Eikonal loss, the properties of the SDF can be well constrained, thereby ensuring the smoothness and continuity of the reconstructed surface.

## 4. Experimentation

### 4.1. Dataset

This paper conducts experimental analysis on the ScanNet [31], Hypersim [32], and Replica [33] datasets, as shown in Table 1. The ScanNet dataset is the main dataset used for indoor scene 3D reconstruction tasks. Our method selects 10 scenes from the ScanNet dataset. For each scene, a set of equally spaced images (about 150–600 images) are sampled from the corresponding video and the images are adjusted to a resolution of 640×480. In addition, a scene is selected from the Hypersim dataset and the Replica dataset to test the generalization of this method in large-scale scenes. The results verify that this method has good reconstruction effects on other datasets in addition to the large public dataset ScanNet.

### 4.2. Comparative Experiment

Five evaluation indicators are used: accuracy (Acc), completeness (Comp), precision (Prec), recall (Recall), and F1 score (F-score).

Accuracy is an indicator to measure the degree of consistency between the reconstructed mesh and the real scene mesh:(13)$$Acc=mean_{p\in P}\left(\underset{p^{*}\in P^{*}}{\min}∥p-p^{*}∥\right)$$
where P represents the set of points in the reconstructed grid, $P^{*}$ is the set of points in the grid of the real scene, p is a point in the set P, and $p^{*}$ is a point in the set $P^{*}$. $∥p-p^{*}∥$ represents the Euclidean distance between P and $P^{*}$. For each point p in P, find the point $p^{*}$ in $P^{*}$ that is closest to it, calculate the distance between them, and average all the distances.

Completeness is used to measure the extent to which the reconstructed model covers the original model or scene:(14)$$Comp=mean_{p^{*}\in P^{*}}\left(\underset{p\in P}{\min}∥p-p^{*}∥\right)$$

This metric measures completeness by calculating the average distance from each point in the true scene mesh $P^{*}$ to the nearest point in the reconstructed mesh P.

Precision is a measure of the proportion of the correctly reconstructed part of the reconstructed model to the entire model. The calculation formula for precision is
(15)$$Prec=mean_{p\in P}\left(\underset{p^{*}\in P^{*}}{\min}∥p-p^{*}∥<0.05\right)$$

For each point p in the reconstructed mesh P, find the point $p^{*}$ in the GT mesh $P^{*}$ that is closest to it, calculate the distance between them, and compare it with the set threshold 0.05 to calculate the proportion less than 0.05.

Recall measures the proportion of points in the GT model that are covered and correctly reconstructed by the reconstruction model:(16)$$Recall=mean_{p^{*}\in P^{*}}\left(\underset{p\in P}{\min}∥p-p^{*}∥<0.05\right)$$

For each point $p^{*}$ in the reconstructed mesh $P^{*}$, find the point p in the GT mesh P that is closest to it, calculate the distance between them, and compare it with the previously set threshold of 0.05 to calculate the proportion less than 0.05.

The F1 score (F-score) is the harmonic mean of precision and recall, which can better measure some unbalanced datasets and provide a more comprehensive evaluation of the overall model:(17)$$F-score=\frac{2\times Prec\times Recall}{Prec+Recall}$$

The architecture of the MLP encoding the SDF in our method consists of eight hidden layers of 256 channels. The training process and related parameters of the model are set as follows: the number of iterations is set to 160,000, the learning rate is set to $4\times 10^{-4}$, the number of rays sampled in each batch of training (batchsize) is set to 1024, and each ray contains 64 coarse sampling points and 64 fine sampling points. In the patchmatching process, the patch size is set to 11×11, the step size is set to 2 for the block matching process, and the NCC threshold ϵ is 0.66. The weights of each loss function, $\lambda_{c}$, $\lambda_{n}$, $\lambda_{e}$ and $\lambda_{d}$, are set to 1.0, 1.0, 0.1, and 0.5, respectively.

As shown in Table 2, the method in this paper is significantly better than the state-of-the art methods, and performs well in most indicators, far exceeding the traditional MVS method and the TSDF-based reconstruction method.

Among the neural implicit reconstruction methods, Neus and VolSDF mainly focus on object-level reconstruction, so they perform poorly on the ScanNet dataset; the first-generation NeRF is mainly used for new perspective synthesis, and no algorithm for extracting grids is proposed. Therefore, direct use of Marching Cubes to extract grids will result in large-scale collapse. This article only draws on many modules of the NeRF, so it is compared here; the I2-SDF method has good results in the synthetic dataset proposed in this article, but it does not have generalization ability and does not work well on the ScanNet dataset with more noise and motion blur.

MonoSDF and Manhattan-SDF are second only to this method in terms of quantitative results. Manhattan-SDF is based on the Manhattan World hypothesis, which assumes that the world is mainly composed of planes aligned with the coordinate axes. It is easily affected by the complexity of indoor scenes. Strict reliance on planes aligned with the coordinate axes for reconstruction may lead to missing or inaccurate details. Therefore, the overall accuracy is lower than this method. MonoSDF has achieved good performance in various indicators, especially accuracy. This is because MonoSDF weakens the reconstruction of complex structures that are difficult to handle during the reconstruction process. Therefore, the reconstructed part is highly overlapped with the scene itself, but some parts of the reconstruction will be lost. Therefore, compared with other evaluation indicators, there is some gap between its recall rate and this method, because the recall rate measures the ratio of the real model that is reconstructed.

In addition to quantitative analysis, this chapter visualizes the reconstruction results and makes qualitative comparisons with the most advanced MonoSDF and Manhattan-SDF. As shown in Figure 6a, compared with GT, this method fills some holes that were not scanned at the time. Compared with Manhattan-SDF, this method has higher accuracy and a smoother reconstruction effect on the surface of objects. Compared with MonoSDF, this method has more accurate reconstruction in many structures (shown by the red dashed line). The specific details in the red dashed box in Figure 6a are shown in Figure 6b. Manhattan-SDF has low reconstruction accuracy. After zooming in on the detail image, larger triangular facets can be seen. From the visualization point of view, the details are far inferior to those of this method. MonoSDF has better overall detail reconstruction—especially, the wall area is relatively smooth—but there are many missing objects. In the right picture of Figure 6b, MonoSDF did not correctly reconstruct the lamp on the table, and only reconstructed a small part of the outline of the objects placed on the bookshelf, while this method restored these details of the real scene well. In the left picture of Figure 6b, MonoSDF and Manhattan-SDF are largely missing chair legs and armrests, while the method in this paper restores the chair structure in the real scene to a large extent.

### 4.3. Ablation Experiment

#### 4.3.1. Normal Geometry Prior Ablation Experiment

This section will demonstrate the effectiveness of the adaptive normal prior scheme added in this paper through quantitative and qualitative ablation experimental analysis. There are three settings in this ablation experiment: (1) no normal prior is added, denoted as w/o N; (2) a normal prior is added, but the adaptive scheme is not used, denoted as w/N, w/o A; (3) the adaptive scheme is used with the normal prior, denoted as Ours. All settings are tested on the ScanNet dataset, Hypersim dataset, and Replica dataset. The evaluation indicators of each setting are shown in Table 3. It can be seen that the reconstruction quality can be significantly improved by adding the geometric prior, because it provides additional geometric constraints and alleviates many ambiguity problems caused by the lack of texture information in the original image. On this basis, the adaptive scheme can remove the incorrectly estimated normal map and further improve the reconstruction quality.

The following will verify the effectiveness of the adaptive normal prior module used in this section through qualitative visualization results. This module provides a certain improvement in the reconstruction of all indoor scenes, but the improvement in thin structure areas and reflective areas is particularly obvious, so these two areas are selected as visualization displays. As shown in Figure 7, a qualitative analysis of the chair leg structure is performed. Figure 7a is the input RGB image, i.e., the reference image, and Figure 7b is the model reconstructed without using the normal prior. The overall reconstruction accuracy is not high, which is reflected in the fact that the reconstruction effect of the surrounding floor and walls is not flat enough, and the surface of the chair is not smooth enough. Figure 7c adds the normal prior, but does not use the adaptive scheme. Although the reconstruction effect of the wall and chair surface is improved, the chair leg part is not reconstructed due to the wrong normal estimation. Figure 7d is the method used in this paper, that is, the adaptive scheme uses the normal prior, which not only greatly improves the accuracy, but also completely reconstructs the thin structure areas such as the chair leg.

In addition to having a good visual improvement effect in thin structure areas, the adaptive normal prior scheme is also effective in some areas due to lighting or specular reflection. The image shown in Figure 8a is a reference image. The sunlight projected from the window forms regular white light and shadows on the ground. These light and shadows usually affect the reconstruction of the flatness of the entire ground because the network will understand them as independent geometric structures. Figure 8b is a model reconstructed without using the normal prior. A more obvious step structure is reconstructed in the light and shadow area, which is not the result of correct reconstruction. The reconstruction granularity in other areas is also obviously insufficient. Figure 8c adds the normal prior but does not use the adaptive scheme. Since most of the normal priors have weakened the erroneous impact of this area on the reconstruction, the erroneous outline of this area is already vague and not as obvious as in Figure 8b. However, there are also a few normal maps closer to the light and shadow area that estimate this area as a geometric structure different from the floor. Therefore, after adding the adaptive scheme, these few normal estimates can be removed, and the reconstruction effect is as smooth as the ground area, as shown in Figure 8d.

Through the above quantitative and qualitative results, it is not difficult to find that the normal prior can significantly improve the reconstruction effect and geometric details in the indoor scene reconstruction task. The use of the adaptive normal prior can reduce the erroneous reconstruction of many geometric structures and regions, and has a good correction effect on thin structures and partially reflective areas, making the reconstruction of these parts more robust and reasonable and close to the real scene, further proving the effectiveness and usability of the adaptive normal prior module in this section.

#### 4.3.2. Ablation Experiment of Distortion Loss Function

The effectiveness of the new regularization term, the distortion loss function, is proposed in this paper through quantitative and qualitative ablation experimental analysis. Since the distortion loss function $L_{dist}$ is only applicable to scenes with floating debris, the ablation experiment is also conducted on these scene datasets. There are two different settings in this ablation experiment: (1) without adding the distortion loss function, denoted as w/o $L_{dist}$; (2) after adding the distortion loss function, denoted as Ours. By ablating it in five scenes with floating debris noise in ScanNet, the quantitative indicators shown in Table 4 can be obtained.

As can be seen from Table 4, Acc and Prec have been significantly improved, which means that the distortion loss function removes some erroneous reconstruction parts and achieves more accurate reconstruction, while Comp and Recall have not changed much, because the method in this paper does not produce too many additional areas for supplementary reconstruction.

Figure 9 shows a visual comparison of a scene with a large amount of floating debris. There are is a large amount of incorrectly reconstructed floating debris in Figure 9a. After adding the distortion loss function, most of this floating debris in Figure 9b is eliminated.

In addition, it also has a good removal effect on single floating debris areas in some scenes. As shown in Figure 10, there are single floating debris areas in both scenes in Figure 10a, which are successfully removed in Figure 10b.

The ablation experiment results on the distortion loss function show that, both qualitatively and quantitatively, the distortion loss function proposed in Chapter 3 of this paper can effectively remove floating object noise in the 3D model and improve the overall quality of the 3D model.

### 4.4. Limitations

The comparative experiment part illustrates that the proposed method has advantages over the benchmark model in both quantitative indicators and visualization results, which fully proves the superiority of the proposed method. The ablation experiment part proves the effectiveness of each module of the proposed method. Nevertheless, the proposed method still has some limitations in some specific scenarios.

For scenes with messy objects, the reconstruction effect of this method is not perfect. This is because in the 3D reconstruction task of indoor scenes, the arrangement and combination of objects will greatly affect the reconstruction process. As shown in Figure 11, there are many irregularly shaped objects on the table in the input image, and they are placed in overlapping and mutually occluding situations. For soft and transparent objects, such as plastic bags, it is difficult to accurately estimate their surface details because their appearance features are not easy to capture. Although this method can reconstruct its general outline, it is difficult to judge that this is a plastic bag based on the outline. The reconstruction of such soft and non-solid objects in the scene has always been a difficult problem, and even the GT model obtained by scanning has not been able to restore this part well.

In addition, the fuzziness of the input image is also crucial to the reconstruction result. If the input image has a certain amount of camera blur, as shown in Figure 5, Figure 6, Figure 7 and Figure 8, many details of the object will be lost in the image, and the loss of these details increases the difficulty of reconstruction. This blur can also lead to inaccurate feature point detection, affecting the feature extraction and feature matching process. This blur can also affect the reconstruction of some areas with insufficient texture features, such as the dividing line and gap between the drawers in Figure 11. Due to the image blur, the area is incorrectly reconstructed.

## 5. Conclusion and Future Work

Based on the implicit expression of the NeRF and SDF, this paper proposes an indoor scene reconstruction method based on adaptive normal priors, and optimizes the geometric learning process through adaptive normal priors. The method proposed in this paper can significantly improve the reconstruction quality of large plane textures and uneven lighting areas in indoor scenes, and can also remove floating debris in the reconstructed 3D model. However, there are still some problems that need to be further optimized and improved in the future:When there are many objects and elements in the scene and they are irregular, this method can only reconstruct the general outline, and the object category cannot be directly determined by these outlines. At the same time, the reconstructed results at the connections between objects and between objects and backgrounds are discontinuous. One solution is to try to introduce more priors to allow the neural network to obtain more useful information to accurately learn and understand the elements in the scene. Another feasible solution is to find a way to distinguish between object areas and non-object areas, and then learn them separately, which is conducive to further capturing more complex details.This method takes several hours to more than ten hours to train and optimize a single indoor scene, which limits the application of this method in reconstruction over a relatively large range and in real-time reconstruction. One possible solution to improve training efficiency is to use hashed multi-resolution encoding to use a smaller network without sacrificing quality, thereby significantly reducing the number of floating-point and memory access operations, allowing the neural network to be trained at a smaller computational cost while maintaining reconstruction quality, greatly reducing training time.

## Figures and Tables

**Figure 1 jimaging-10-00231-f001:**
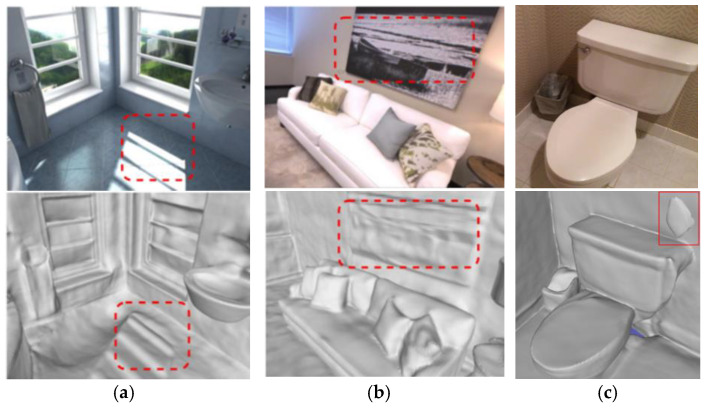
(**a**) Distortion of reconstructed 3D models under uneven lighting conditions enclosed by the red dashed box; (**b**) distortion of 3D reconstruction of smooth planar texture areas enclosed by the red dashed box; (**c**) floating debris noise in red box in 3D reconstruction.

**Figure 2 jimaging-10-00231-f002:**
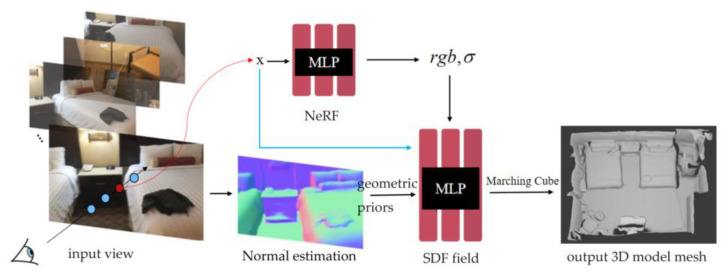
Overall framework of the method.

**Figure 3 jimaging-10-00231-f003:**
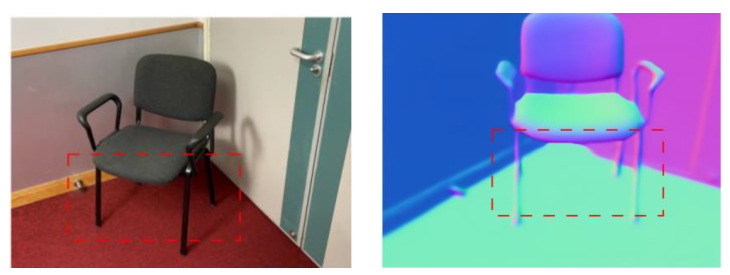
(**a**) The normal estimation is inaccurate in some fine structures enclosed by red dashed box, such as chair legs, based on TiltedSN normal estimation module; (**b**) we use an adaptive normal prior method to derive accurate normals based on the consistency of adjacent images. In the red dashed box, the fine structures are accurately reconstructed.

**Figure 4 jimaging-10-00231-f004:**
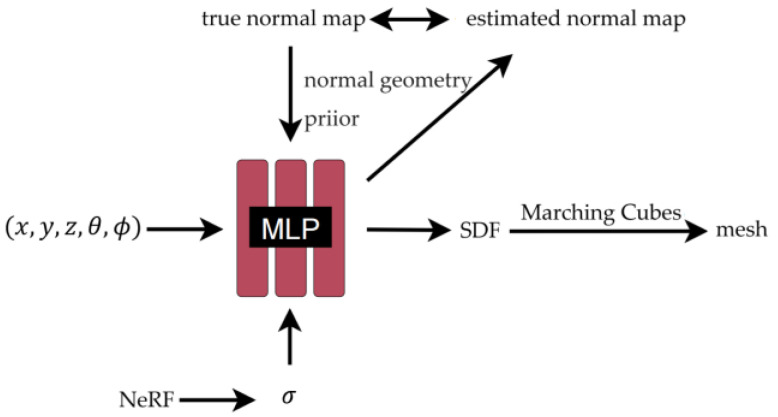
Neural implicit reconstruction process.

**Figure 5 jimaging-10-00231-f005:**
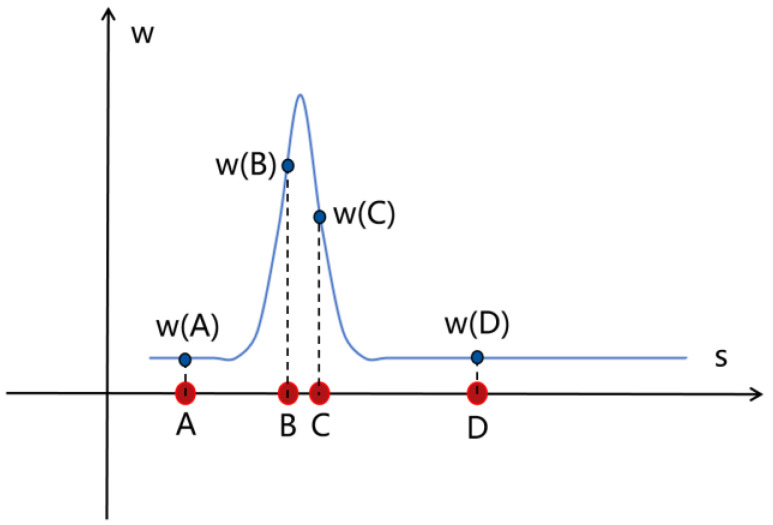
Distribution diagram of distance and weight values between sampling points.

**Figure 6 jimaging-10-00231-f006:**
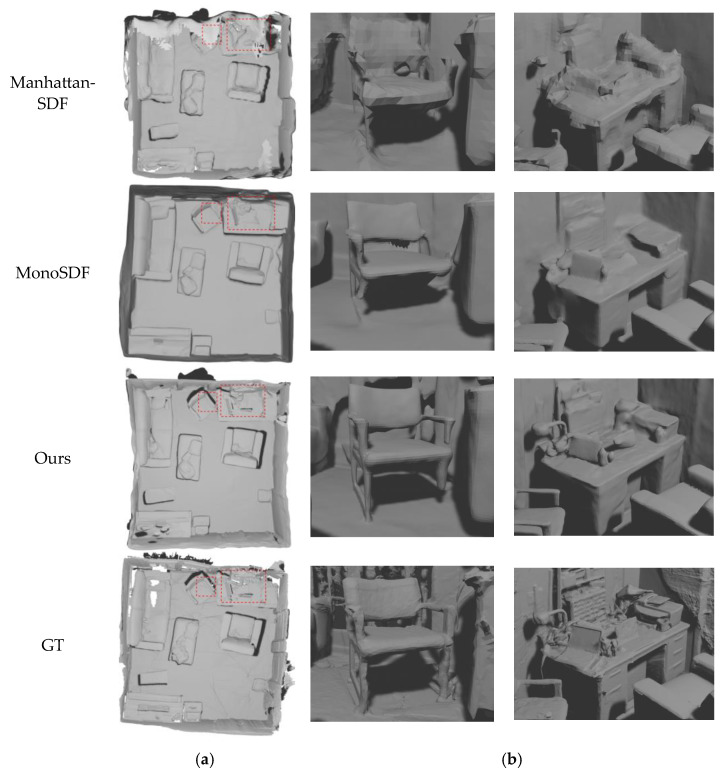
Three-dimensional model reconstructed from scenes in the ScanNet dataset. (**a**) Comparison of 3D models; (**b**) comparison of the specific details in the red dashed box.

**Figure 7 jimaging-10-00231-f007:**
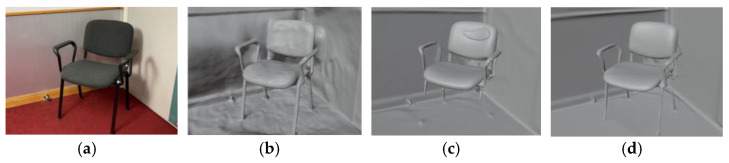
Qualitive comparison for thin structure areas using ScanNet dataset: (**a**) reference image; (**b**) model reconstructed without using normal prior; (**c**) model reconstructed with normal prior and without adaptive scheme; (**d**) model reconstructed with normal prior and adaptive scheme.

**Figure 8 jimaging-10-00231-f008:**
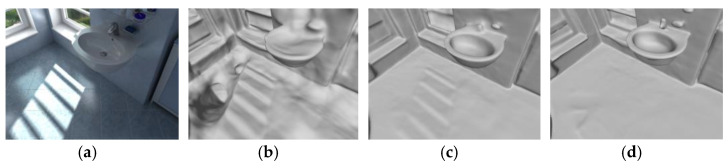
Qualitive comparison for reflective areas using Hypersim dataset: (**a**) reference image; (**b**) model reconstructed without using normal prior; (**c**) model reconstructed with normal prior and without adaptive scheme; (**d**) model reconstructed with normal prior and adaptive scheme.

**Figure 9 jimaging-10-00231-f009:**
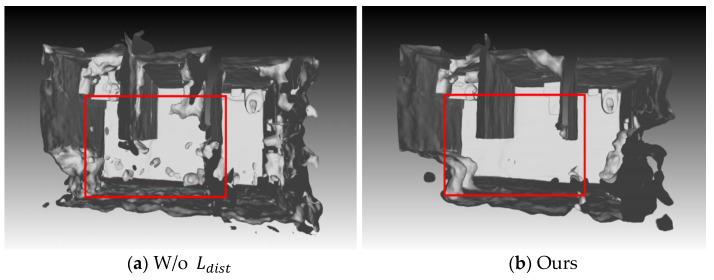
Visual comparison for a scene with a large amount of floating debris using the ScanNet dataset; (**a**) reconstruction result without adding a distortion loss function; (**b**) reconstruction result with a distortion loss function.

**Figure 10 jimaging-10-00231-f010:**
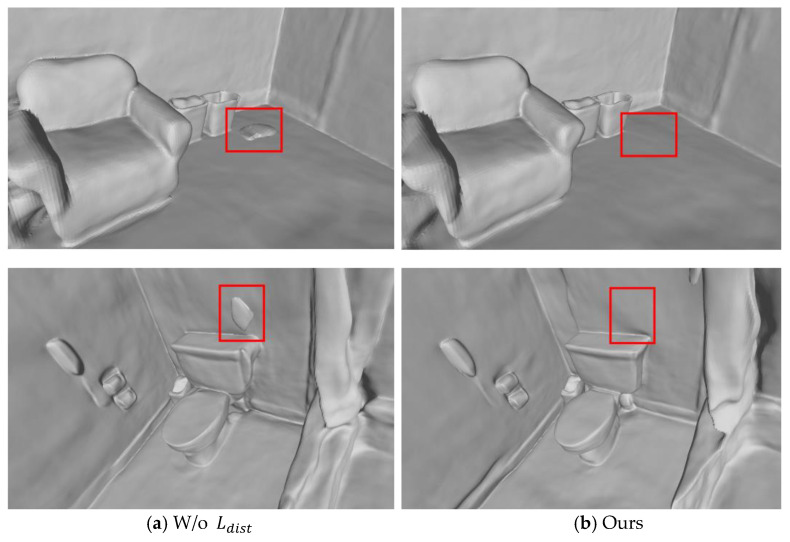
Visual comparison for a scene with single floating debris areas enclosed by red dashed box using the ScanNet dataset; (**a**) reconstruction result without adding distortion loss function; (**b**) reconstruction result with distortion loss function.

**Figure 11 jimaging-10-00231-f011:**
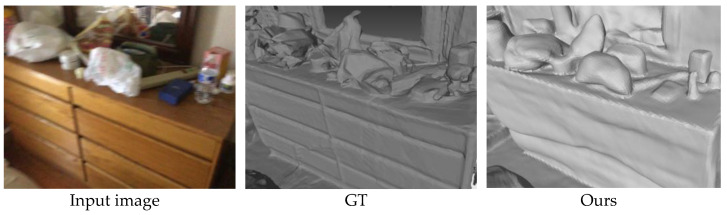
The limitations of this method in the 3D reconstruction of scenes with clutter, occlusion, soft non-solid objects, and blurred images, using the ScanNet dataset.

**Table 1 jimaging-10-00231-t001:** Selected datasets to test our method.

Dataset	Scene Number	Scenes Selected in This Paper
ScanNet	1500+	10
Hypersim	461	10
Replica	18	10

**Table 2 jimaging-10-00231-t002:** Quantitative comparison with other methods.

Method	Acc ↓	Comp ↓	Prec ↑	Recall ↑	F-Score ↑
COLMAP [10]	0.047	0.235	0.711	0.441	0.537
Atlas [8]	0.211	0.070	0.500	0.659	0.564
NeuralRacon [11]	0.056	0.081	0.545	0.604	0.572
Neus [24]	0.179	0.208	0.313	0.275	0.291
VolSDF [25]	0.414	0.120	0.321	0.394	0.346
NeRF [34]	0.735	0.177	0.131	0.290	0.176
Manhattan-SDF [35]	0.072	0.068	0.621	0.586	0.602
MonoSDF [36]	**0.035**	**0.048**	0.799	0.681	0.733
I2-SDF [37]	0.066	0.070	0.605	0.575	0.590
Ours	0.037	**0.048**	**0.801**	**0.702**	**0.748**

Bold text indicates the best results and the underlined text indicates the second best results. ↓ indicates that the smaller the value of this indicator, the better; ↑ indicates that the larger the value of this indicator, the better.

**Table 3 jimaging-10-00231-t003:** Normal geometry prior ablation experiment.

Method	Acc ↓	Comp ↓	Prec ↑	Recall ↑	F-Score ↑
w/o N	0.183	0.152	0.286	0.290	0.284
w/N, w/o A	0.050	0.053	0.759	0.699	0.727
Ours	**0.037**	**0.048**	**0.805**	**0.709**	**0.753**

Bold text indicates the best results. ↓ indicates that the smaller the value of this indicator, the better; ↑ indicates that the larger the value of this indicator, the better.

**Table 4 jimaging-10-00231-t004:** Distortion loss function ablation experiment.

Method	Acc ↓	Comp ↓	Prec ↑	Recall ↑	F-Score ↑
w/o $L_{dist}$	0.055	**0.052**	0.742	0.701	0.721
Ours	**0.047**	**0.052**	**0.795**	**0.704**	**0.746**

Bold text indicates the best results. ↓ indicates that the smaller the value of this indicator, the better; ↑ indicates that the larger the value of this indicator, the better.

## Data Availability

The dataset used in this study is open access and can be found here: ScanNet dataset: http://www.scan-net.org/, accessed on 4 July 2024; Hypersim dataset: https://github.com/apple/ml-hypersim, accessed on 4 July 2024; Replica dataset: https://github.com/facebookresearch/Replica-Dataset/releases, accessed on 4 July 2024.

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
