# Peer review of "Three-Dimensional Reconstruction of Indoor Scenes Based on Implicit Neural Representation"

_2313-433X, 2024, doi:10.3390/jimaging10090231_

Round 1
Reviewer 1 Report
Comments and Suggestions for Authors
3D reconstruction of indoor scenes based on neural implicit
The paper written well based on achieved results, I have some minor revisions that may help improve paper's quality
1. The abstract looks shorter than traditional scientific papers, try to extant it based on main contributions.
2. Figure 2. Overall framework of the method. The authors should explain it in detail since it is the main flowchart of the proposed method.
3. Dataset. Please make a Table based on ScanNet[21], Hypersim[22] and 361 Replica [23] datasets and show us how many images they have and you used.
4. The Conclusion section also short as Abstract, should be included faced problems during the training and testing process and future directions to solve that isuues.
Comments on the Quality of English Languageok
Author Response
Comments 1: The abstract looks shorter than traditional scientific papers, try to extant it based on main contributions.
Response 1: Thank you for pointing this out. I agree with this comment. Therefore I’ve extended the abstract with more detailed information about our contributions.
Comments 2: Figure 2. Overall framework of the method. The authors should explain it in detail since it is the main flowchart of the proposed method.
Response 2: Thank you for pointing this out. I agree with this comment. Brief explanation about each modules in this method framework are provided.These new edits can be found from line 175 -192.
Comments 3: Dataset. Please make a Table based on ScanNet[21], Hypersim[22] and 361 Replica [23] datasets and show us how many images they have and you used.
Response 3: Thank you for pointing this out. I agree with this comment. Information about these 3 datasets and how we use these datasets is addressed and also a table is given to visualize above description.these new edits can be found from line 415-425.
Comments 4: The Conclusion section also short as Abstract, should be included faced problems during the training and testing process and future directions to solve that issues.
Response 4: Thank you for pointing this out. I agree with this comment. The Conclusion section has been reworded with more detail and the faced problems and future directions to solve those problems are introduced.these new edits can be found from line 631-654.
Reviewer 2 Report
Comments and Suggestions for Authors
3D Reconstruction of Indoor Scenes Based on Neural Implicit Representations
The current manuscript presents an innovative and comprehensive approach to 3D reconstruction of indoor scenes using neural implicit representations, specifically combining neural radiance fields (NeRF) and signed distance functions (SDF). This method addresses key challenges such as handling planar textures, uneven lighting, and floating debris, which have been persistent issues in the field. The proposed approach significantly improves reconstruction quality, as demonstrated through extensive experiments. However, a few aspects of the paper could be further improved to enhance clarity and presentation. Below are the combined suggestions for improving the manuscript:
1. Figure-Related Suggestions:
- Font Consistency: Ensure that the fonts used in all figures are uniform throughout the manuscript for a more professional and visually coherent presentation.
- Resolution Adjustment: Several figures appear to have low resolution, which affects clarity. Adjusting these figures to higher resolution would improve readability and presentation quality.
2. Extension of Applications:
- Discussion of Generalization: Although the focus is on indoor scenes, it would be beneficial to briefly discuss how the proposed method could be extended to outdoor or larger-scale environments. This would broaden the potential applications of the research and attract interest from a wider audience in the computer vision community.
3. Conclusion and Future Directions:
- Outline of Future Work: Given the promising results, adding a section discussing potential future research directions would strengthen the conclusion. For example, exploring how this method could be adapted for dynamic or more complex environments would highlight the method’s flexibility and potential long-term impact.
Author Response
Comments 1:Figure-Related Suggestions:
Font Consistency: Ensure that the fonts used in all figures are uniform throughout the manuscript for a more professional and visually coherent presentation.
Resolution Adjustment: Several figures appear to have low resolution, which affects clarity. Adjusting these figures to higher resolution would improve readability and presentation quality.
Response 1: Thank you for pointing this out. I agree with this comment.The fonts and sizes in the pictures have been unified, and low-resolution pictures have been replaced with higher-resolution pictures.
Comments 2:Extension of Applications:
Discussion of Generalization: Although the focus is on indoor scenes, it would be beneficial to briefly discuss how the proposed method could be extended to outdoor or larger-scale environments. This would broaden the potential applications of the research and attract interest from a wider audience in the computer vision community.
Response 2:Thank you for pointing out this issue. I would like to make the following clarification regarding this opinion: This paper mainly solves the problems of large planar textures, uneven lighting, and floating debris in indoor scenes. These problems also exist in outdoor scenes, so this solution can also be used for 3D reconstruction of outdoor scenes. However, outdoor scenes also have many unique difficulties, such as complex textures, large shadows, moving objects, etc. This solution mainly studies the 3D reconstruction of indoor scenes, so it does not involve the research of these problems.
Comments 3: Conclusion and Future Directions:
Outline of Future Work: Given the promising results, adding a section discussing potential future research directions would strengthen the conclusion. For example, exploring how this method could be adapted for dynamic or more complex environments would highlight the method’s flexibility and potential long-term impact.
Response 3: Thank you for pointing this out. I agree with this comment. We add more contents to the conclusion section to address what kind of scenarios that our method may well adaptive to and future directions that may solve the existing problem. these new edits can be found from line 631-654.
Reviewer 3 Report
Comments and Suggestions for Authors
The article presents an methodology for indoor scene 3D reconstruction from 2D images using multilayer perceptron. It also includes debris removal to construct a high quality 3D mesh model. The methodology is clearly explained, different experiments were carried out as well as comparison with other methods is performed. The proposed approach is rather limited in cases if small objects or objects containing small surface details are presented for reconstruction. The article demonstrates the importance of normal priors to get the better quality. The article is well written and structured, but the Related Works section could be improved. Please consider the following comments:
- The Related Works section contains only 17 references and should be expanded to better represent the current state of the art. The related works are grouped to only 3D reconstructions based on MVS or neural implicit and are limited by indoor scenes. The methods could be also analyzed from other perspectives (e.g. input/output type (mesh, cubes, ...)) or clearly stating the benefits and weak points of one approach versus another.
- (17 line) Abbreviation SDF is defined only in 198 line (it should be defined with the first use).
- (Figures 6-11) Please state what dataset was used in each figure.
- (4 Experimentation section (360 line)) Please describe the architecture of the multilayer perceptron and the training procedure (training parameters, what data was used for training /testing, what technical equipment was used for training, and similar).
- (366 line) The authors state that "five evaluation indicators are used: accuracy (Acc), completeness (Comp), precision (Prec), recall (Recall) and F1 score (F-score)". Please provide the formulas which were used to calculate these values and the procedure.
- (4.2 section, 366 line) Have you considered to compare the methods by the means of time and calculations / memory required? Compared to other methods, does Your methods work faster, slower or does not have significant difference?
- (Conclusions section) The section only reviews the proposed methodology rather than presents the actual findings. This section should state the benefits of the proposed methodology, including numerical evaluation.
Author Response
Comments 1:- The Related Works section contains only 17 references and should be expanded to better represent the current state of the art. The related works are grouped to only 3D reconstructions based on MVS or neural implicit and are limited by indoor scenes. The methods could be also analyzed from other perspectives (e.g. input/output type (mesh, cubes, ...)) or clearly stating the benefits and weak points of one approach versus another.
Response 1: Thank you for your kind suggestions. In the Related work section, in addition to the original analysis of 3D reconstruction technology based on MVS and neural implicit, more comparative technology analysis has been added, such as the comparative analysis of indoor 3D reconstruction technology based on visual SLAM and based on TSDF(please see the edits in line 66-102), and the number of references has also been expanded.
Comments 2: - (17 line) Abbreviation SDF is defined only in 198 line (it should be defined with the first use).
Response 2: Thank you for pointing out this issue. I have also made unified modifications to other similar issues. SDF and NeRF appears in the Abstract for the first time, so their abbreviations are defined in the abstract.
Comment 3: - (Figures 6-11) Please state what dataset was used in each figure.
Response 3: Thank you for pointing this out. I agree with your suggestion. In each figure from 6-11, which dataset being used was now explicitly addressed.
Comments 4: - (4 Experimentation section (360 line)) Please describe the architecture of the multilayer perceptron and the training procedure (training parameters, what data was used for training /testing, what technical equipment was used for training, and similar).
Response 4: Thank you for pointing this out. I agree with your comment. The basic structure of the MLP used in this method is supplemented, including the number of hidden layers and channels, as well as the training process and related parameter settings. These new edits can be found from line 458-464.
Comments 5: - (366 line) The authors state that "five evaluation indicators are used: accuracy (Acc), completeness (Comp), precision (Prec), recall (Recall) and F1 score (F-score)". Please provide the formulas which were used to calculate these values and the procedure.
Response 5: Thank you for this comment. The formulas for these five indicators are added as requested and short description is given as well. These new edits can be found from line 428-457.
Comments 6: - (4.2 section, 366 line) Have you considered to compare the methods by the means of time and calculations / memory required? Compared to other methods, does Your methods work faster, slower or does not have significant difference?
Response 6: Thank you for your question. I’d like to provide some short clarification on the running time of our method in the conclusion section, and some possible improvements are given as well. These new edits can be found from line 647-654.
Comments 7: - (Conclusions section) The section only reviews the proposed methodology rather than presents the actual findings. This section should state the benefits of the proposed methodology, including numerical evaluation.
Response 7:Thank you for your comments. The conclusion section has been improved, briefly summarizing the advantages of the proposed method, describing the existing problems, and further providing some ideas for improving these problems in the future. These new edits can be found from line 631-654.